# Supramolecular networks stabilise and functionalise black phosphorus

Vladimir V. Korolkov [1], Ivan G. Timokhin[2], Rolf Haubrichs[2], Emily F. Smith[3], Lixu Yang[3], Sihai Yang[4], Neil R. Champness [3], Martin Schröder[4] & Peter H. Beton [1]

The limited stability of the surface of black phosphorus (BP) under atmospheric conditions is a significant constraint on the exploitation of this layered material and its few layer analogue, phosphorene, as an optoelectronic material. Here we show that supramolecular networks stabilised by hydrogen bonding can be formed on BP, and that these monolayer-thick films can passivate the BP surface and inhibit oxidation under ambient conditions. The supramolecular layers are formed by solution deposition and we use atomic force microscopy to obtain images of the BP surface and hexagonal supramolecular networks of trimesic acid and melamine cyanurate (CA.M) under ambient conditions. The CA.M network is aligned with rows of phosphorus atoms and forms large domains which passivate the BP surface for more than a month, and also provides a stable supramolecular platform for the sequential deposition of 1,2,4,5-tetrakis(4-carboxyphenyl)benzene to form supramolecular heterostructures.

[1] School of Physics & Astronomy, University of Nottingham, Nottingham NG7 2RD, UK. [2] CristalTech Sàrl, Rue du Pré-Bouvier 7, CH-1242 Satigny, Switzerland. [3] School of Chemistry, University of Nottingham, Nottingham NG7 2RD, UK. [4] School of Chemistry, The University of Manchester, Oxford Road, Manchester M13 9PL, UK. Correspondence and requests for materials should be addressed to V.V.K. (email: vladimir.korolkov@nottingham.ac.uk) or to P.H.B. (email: peter.beton@nottingham.ac.uk)

Black phosphorus (BP), one of the several allotropic forms of phosphorus, has a layered structure and is a narrow gap semiconductor with a bulk band gap of ~0.3 eV[1–3]. Similar to other layered materials it can be exfoliated[4–6] with scotch tape to form a single layer of BP known as phosphorene[5]. Unlike gapless graphene[7], phosphorene has a band gap which was predicted, and later confirmed to be ~2 eV[8–10]. The band gap is thickness dependent and thus can be readily tuned[9, 11]. Since the first report of exfoliation of BP[4], and some 100 years after the first high-pressure synthesis of BP crystals by Bridgman in 1914[12, 13], phosphorene or few-layered BP has been used widely to fabricate transistors[4, 14–16], including flexible devices[17, 18]. In addition, van der Waals heterostructures[19] based on phosphorene and transition metal dichalcogenides show promising optoelectronic properties[20], phosphorene–graphene hybrid materials have been proposed as high-capacity anodes for sodium-ion batteries[21] and the number of possible applications is growing rapidly[22].

One of the biggest challenges in BP and phosphorene research remains its stability under atmospheric conditions. As previously reported, the surface of BP is prone to oxidation upon exposure to the ambient environment[23], a particular problem for monolayer-thick phosphorene[24]. The exact mechanism of oxidation is not yet fully understood, but the presence of water and light along with oxygen facilitate surface degradation with the formation of a species of oxidised phosphorus[23, 25]. This has led to several approaches to passivate the surface of both BP and phosphorene including the growth of $Al_2O_3$ on BP[24], covalent attachment of phenyl groups via diazonium chemistry[26], stabilisation in specific solvents[27], and the encapsulation of phosphorene between layers of hexagonal boron nitride (hBN)[28–30]. The recent report of the successful passivation of BP via the covalent attachment of p-nitrophenyl and p-methoxyphenyl groups[26] is promising. Nevertheless, complementary routes in which the protective layer interacts with the BP via noncovalent interactions are also of great interest, as demonstrated, for example, by recent studies of the adsorption of 7,7,8,8-tetracyano-p-quinodimethane and various perylene derivatives on BP flakes produced by liquid-phase exfoliation[31, 32]. The use of noncovalently bound overlayers is particularly relevant to phosphorene, for which covalent linkages are likely to modify strongly the electronic structure. In addition, the use of thick encapsulation layers to form a buried interface, while effective, is not readily compatible with the exploration and exploitation of the potentially interesting interactions between the BP surface and its environment.

In this paper we explore a new route to the solution of this problem through an investigation of the compatibility of BP with the formation of supramolecular networks which have monolayer thickness and are stabilised by noncovalent in-plane interactions, specifically hydrogen bonding. This is motivated by the extensive literature[33, 34] of the formation of similar supramolecular networks on graphite, which has some structural similarity to BP but is much less reactive. We demonstrate that supramolecular networks can be formed on BP by depositing a mono-component nanoporous array of trimesic acid (TMA), and the bimolecular network formed by cyanuric acid (CA) and melamine (M). While the more open TMA array does not passivate the BP surface, the hexagonal melamine cyanurate (CA.M) array is highly effective and provides protection under ambient conditions over a period of more than a month. In addition, we identify the orientation of the CA.M relative to the rows of phosphorus atoms at the surface and, normal to the rows, observe moiré effects which are characteristic of a well-ordered interfacial structure. We have also demonstrated that CA.M monolayers on BP provide a stable platform for the sequential growth of additional molecular layers, for example, 1,2,4,5-tetrakis(4-carboxyphenyl)benzene (TCPB), leading to the formation of a supramolecular heterostructure, and

demonstrating the facility for further functionalisation of the BP substrate.

## Results

**Atomic force microscopy of BP.** For this study we have used crystals of BP grown by CristalTech Sàrl (Satigny, Switzerland) which were synthesised according to the protocol developed by Lange et al.[35]. The surface of BP was prepared prior to molecular deposition by removal of the top few layers using scotch tape; our experiments are thus performed on the surface of a thick, bulk-like, BP crystal. This procedure was carried out under ambient conditions at room temperature and under standard laboratory illumination.

Under atmospheric pressure, and at room temperature, BP has an orthorhombic layered structure. Within a single layer, phosphorus atoms are organised in a hexagonal arrangement but, in contrast to hexagonal boron nitride and graphene, the hexagonal motif in BP is puckered reflecting the sp³ character of the hybridisation of the P centres[36] (see Fig. 1a). Each P atom thus forms three covalent bonds with neighbouring P atoms and a fourth orbital carries a lone pair of electrons. This lone pair is responsible for the higher reactivity of the BP surface as compared with graphite/graphene and hBN.

An image of a freshly cleaved BP surface (Fig. 1c) acquired using atomic force microscopy (AFM; see Methods) shows typical features such as surface cracks of various depths, residual nanoflakes (marked 1 and 2) and large areas of atomically flat surface. We found that it is possible to scan the surface for 2–3 h under ambient conditions after exfoliation before it starts to degrade due to oxidation. This is sufficiently long to determine the essential topographic information and some details of the atomic structure of the surface, such as the parallel lines with a separation of $0.43 \pm 0.02$ nm which are resolved in Fig. 1b. This

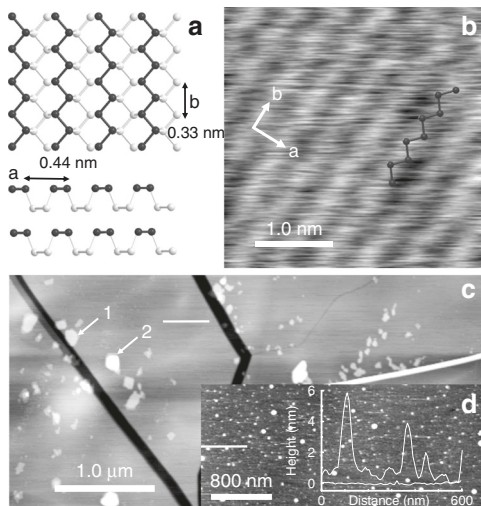

**Fig. 1** Overview of the black phosphorus surface. **a** Schematic showing the surface structure of black phosphorus. **b** Tapping mode AFM showing the surface lattice structure with an overlay representing the surface phosphorus atoms. **c** Large-scale AFM image showing structural features on the freshly cleaved surface of a single crystal of BP. Islands marked 1 and 2 have heights of $3.35 \pm 0.1$ nm and $9.6 \pm 0.1$ nm, respectively. Both images were acquired with a NuNano probe with a second harmonic of 2.385 MHz. **d** AFM scan of the same sample of BP taken 2 days later. An inset shows two surface profiles taken along the white lines in Fig. 1c, d which highlight the enhanced roughness after prolonged exposure to the atmosphere due to oxidation. This image was acquired with a Multi75Al-G probe using the fundamental resonance of 69.46 kHz

value is very close to the previously reported surface lattice constant[37] of BP. In common with published studies of BP using scanning tunnelling microscopy[23, 37] we attribute these rows to the upper P atoms marked in bold in Fig. 1a as highlighted by the overlaid zigzag schematic of P atoms in Fig. 1b. Surface oxidation prevented the acquisition of higher-resolution images; however, high resolution was obtained following the deposition of the supramolecular layers as discussed below.

A typical AFM scan of BP exposed to the air for 2 days is shown in Fig. 1d. In this image the step edges are no longer resolved and the surface roughness has increased by an order of magnitude from ~0.2 nm in Fig. 1c to ~2 nm in Fig. 1d. Typical profiles from these images are shown as an inset to Fig. 1d. It was not possible to form supramolecular arrays on such a surface.

**Molecular adsorption on BP.** For our investigations of the formation of hydrogen-bonded supramolecular arrays we have chosen TMA[38–41], M[42–45] and CA[44, 46], all of which are known to form supramolecular arrays on a number of substrates and have been extensively studied over the past two decades. All molecules were deposited from solution under ambient conditions (see Methods).

The molecular structure of TMA is shown in Fig. 2a. Our AFM images (Fig. 2b, c) reveal that, following the immersion of BP in a solution of TMA in EtOH for 180 s, islands of TMA are deposited on the surface. Under these conditions the island size did not normally exceed ~100 nm and profiles (Fig. 2e) extracted from Fig. 2b show a step height of ~0.4 nm consistent with a monolayer thickness. The overall surface coverage is ~0.3–0.4 monolayers, but a precise estimate was complicated by the deterioration of the surface, presumably due to oxidation, within the first 2–3 h of scanning. Higher-resolution images of the islands revealed a honeycomb arrangement of TMA molecules with a period of 1.7 nm (Fig. 2c), which we identify as the nanoporous 'chickenwire'

structure (see schematic in Fig. 2a), in which each TMA molecule forms six in-plane hydrogen bonds with neighbouring molecules[47]. This is one of several arrangements which has been reported for TMA monolayers on graphite and metals under ambient, liquid and vacuum conditions[39–41]. We did not observe any other molecular phases.

The results (Fig. 2) show conclusively that supramolecular layers of TMA can be formed on BP, but the surface is not significantly passivated by this termination. In fact, more detailed examination of, for example, the relative orientation of the molecular structure in relation to the atomic surface lattice proved to be difficult due to the rapid degradation of the surface which, as revealed by AFM, starts on the uncoated areas of the surface (Fig. 2d) suggesting that TMA islands provide only a temporary protection of the surface.

We have also investigated the adsorption of melamine cyanurate, CA.M, a well-studied supramolecular system that has been deposited on a range of substrates[44, 48–52] and forms a hexagonal network in which each melamine molecule forms nine hydrogen bonds with three cyanuric acid molecules (Fig. 3b). We have recently observed[53] that when deposited on hBN, CA.M provides a platform that facilitates further molecular adsorption from a range of solvents to form supramolecular hetereostructures.

We deposited CA.M by immersion in an aqueous solution (see Methods) using a similar protocol[53] that leads to a near-monolayer coverage on hBN. High-resolution AFM images reveal the expected honeycomb array of CA.M (Fig. 3a, b) with a measured period of $a_{CA.M} = 0.99 \pm 0.02$ nm which agrees well with previous studies[44, 46, 48]. Figure 3a also reveals a linear moiré pattern with a period of $9.1 \pm 0.5$ nm (Fig. 3a) which is aligned with one of the principal axes of the molecular network.

If the tip-sample force is progressively increased when imaging in contact mode, the monolayer CA.M may be locally removed (the threshold force is estimated to be ~2–3 nN). This procedure facilitates the acquisition of contact mode AFM images of the underlying BP surface which, as shown in the Fig. 3a overlay, reveal rows with a separation of $0.44 \pm 0.02$ nm, which we attribute to the zigzag chains of top-layer phosphorus atoms. Also resolved along the rows are periodic topographic features with a spacing of $0.33 \pm 0.02$ nm; this value matches closely the periodicity along the zigzag chains (the vector **b** in Fig. 1a)

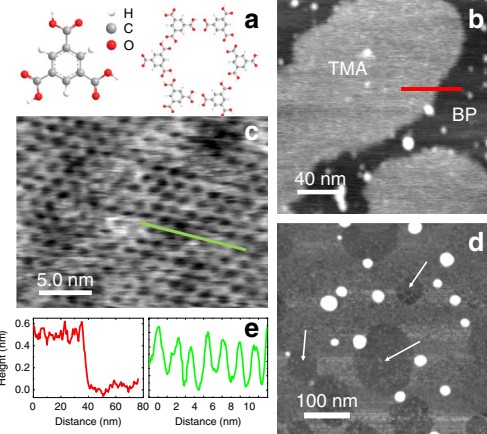

**Fig. 2** Assembly of trimesic acid on black phosphorus. **a** Structures of trimesic acid (TMA) and the honeycomb structural motif. **b** AFM scan of two islands of TMA acquired in tapping mode using the fundamental resonance of 59.97 kHz; the sample was prepared by immersion of BP for 180 s in a 190 μM solution of TMA in EtOH. **c** High-resolution AFM scan showing honeycomb arrangement of TMA within the island on scan **b**. The scan was acquired in tapping using the second harmonic with resonant frequency of 390.8 kHz. **d** A large area AFM image taken 4 h after the sample was prepared by immersion for 60 s of BP in a 190 μM solution of TMA in EtOH. The arrows show the location of TMA islands; these appear as depressions since the surrounding BP has been partially oxidised. The scan was acquired in tapping mode using the fundamental resonance with resonant frequency of 61.63 kHz. **e** A cross-section showing the height of the island on scan **b** (~0.4 nm) and the TMA periodic structure on scan **c**

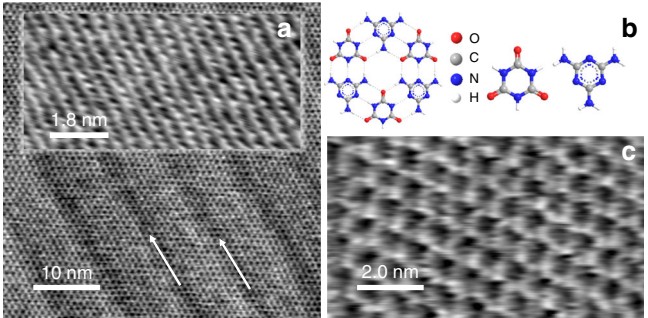

**Fig. 3** Melamine cyanurate deposited on the black phosphorus surface. **a** High-resolution tapping mode AFM scans of melamine cyanurate (CA.M) and the underlying surface of black phosphorus. Arrows show the direction of the CA.M principal axis which is aligned with the BP rows. The CA.M structure was resolved with Multi75Al-G probe using the third harmonic at 1.171 MHz. The image of the surface lattice was acquired in contact mode with an Arrow UHF probe. **b** Schematic of melamine and cyanuric acid and the CA.M complex. **c** High-resolution contact mode AFM scan showing CA.M structure where individual molecules are resolved. The image was acquired with Arrow UHF probe

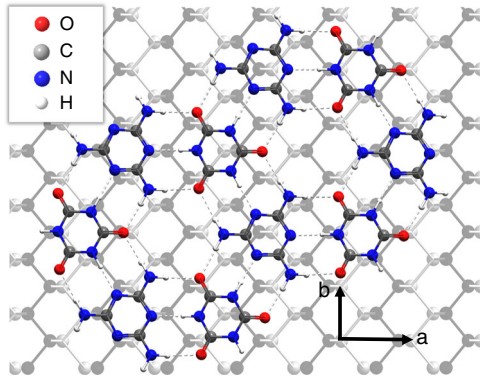

**Fig. 4** Diagram showing the relative orientation of melamine cyanurate on the surface of black phosphorus. Vectors **a** and **b** show orientation of the substrate

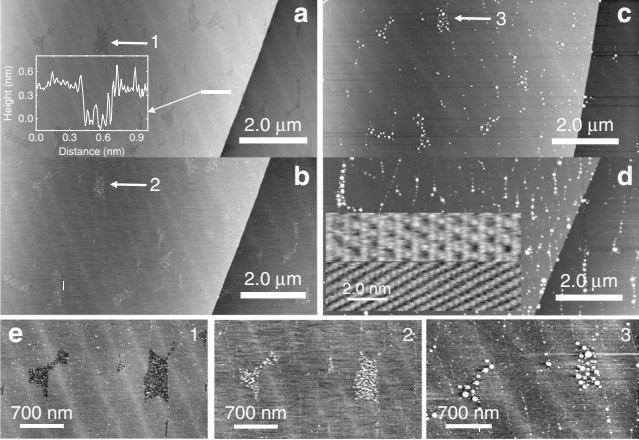

**Fig. 5** Exposure of the melamine cyanurate-terminated black phosphorus surface to atmosphere. **a**–**c** A sequence of tapping mode AFM scans of the same area showing partial oxidation on uncoated areas of the surface. Scans **b**, **c** and **d** were collected 4 h 35 m, 52 h and 35 days, respectively, after sample preparation. A high-resolution inset on image **d** was acquired 1 month after sample preparation with a Multi75Al-G probe using the third harmonic of 1205 kHz. The upper region of the inset shows the intact CA.M network whereas the lower part shows an area where the CA.M layer has been removed to expose the BP surface (as in Fig. 3a); the P rows are clearly visible confirming that no oxidation has occurred. Additional images showing this effect at different times, and over larger areas are included in Supplementary Information (Supplementary Figs. 1–3). **e** 1–3 show the same areas marked 1–3 on scans **a**–**c**. All large-scale AFM scans were acquired with Multi75Al-G cantilever using the fundamental resonance of 64.54 kHz

which was previously determined using scanning tunnelling microscopy[37], in bulk[54] and calculated using density functional theory[37]. The observation of these chains provides strong evidence that the interfacial layer of phosphorus atoms remains intact and is not oxidised.

The relative alignment of the BP rows and the supramolecular overlayer may be determined following the removal of the CA.M monolayer; we find that one of the principal axes of CA.M is aligned with zigzag chains of phosphorus atoms (see schematic in Fig. 4). We note the match, within experimental error, between the CA.M period and 3 times the BP lattice constant along the rows, **b**. It is likely that this commensurability guides the alignment of the BP rows and the CA.M. We also emphasise that the BP represents a substrate with lower, uniaxial, symmetry than hexagonal materials such as hBN or graphite which are commonly used to investigate supramolecular assembly on

surfaces. Interestingly, we observe only one orientation of CA. M domains which is consistent with this lower symmetry and a commensurability driven alignment. Although there is no simple commensurability between the separation of BP rows (lattice vector **b**) and the molecular dimensions, the presence of a moiré pattern of stripes confirms that the CA.M/BP interface is highly ordered.

**Protective effect of melamine cyanurate**. The molecular layer of CA.M is not significantly affected by continuous exposure to the atmosphere. Indeed, the sample discussed in Fig. 3a had been exposed to ambient conditions for 2 days prior to removal of the CA.M layer implying that the presence of this molecular monolayer inhibits the degradation of the underlying BP. The protective effect of CA.M is presented in more detail in Fig. 5. AFM scans (10 μm × 10 μm) of a specific area on the CA.M-terminated BP were acquired over a period of 35 days. During this period the sample was left in an unsealed sample storage box and no measures were taken to prevent it interacting with the ambient environment. The scans include a terrace step in the BP to aid registration.

Immediately after deposition of CA.M (Fig. 5a) we find that ~97% of the surface area is terminated by the molecular layer, and the voids in the coverage appear as dark contrast regions, one of which is marked by an arrow in Fig. 5a. A profile across one such depression shows a height of $0.35 \pm 0.02$ nm consistent with a monolayer (Fig. 5a inset) thickness of CA.M. A zoom of the area marked by an arrow is shown in Fig. 5e (marked 1). We attribute the darker contrast region to the exposed unterminated BP surface; note also that the roughness across the molecular layer and the exposed BP is similar and, in both cases, is in the sub-nanometre range.

Subsequent scans of the same area are shown in Fig. 5b, c (Fig. 5d was taken from the same area but displaced slightly due to damage in sample handling). It is clear that the surface roughness increases on the exposed BP over time due to environmental degradation. For example, progressive roughening of the exposed BP is shown in Fig. 1e; panels 2 and 3 represent subsequent scans of the same area (zooms of the areas identified by arrows in Fig. 5b, c respectively). Figure 5e clearly shows that the depression in the initial surface (1) evolves into a topographically high feature due to oxidation (panels 2 and 3). The roughness is ~25 nm for Fig. 5d, while the initial value is in the sub-nanometre range. Nevertheless, the CA.M survives this exposure as shown in the inset to Fig. 5d; these images are acquired 1 month after sample preparation and show the molecular network (upper part of inset) and the same region after removing the molecular layer using the protocol discussed above in relation to Fig. 3a. This inset shows clearly that the rows of phosphorus atoms can still be clearly resolved in the regions of the surface which have been capped by the CA.M network in contrast to exposed, uncapped regions of the surface. This confirms that the CA.M monolayer termination of BP protects the surface from atmospheric degradation for periods of over a month. Additional images (including similar tests at intermediate intervals) are included in the Supplementary Information (Supplementary Figs. 1–3).

The passivation effect is also effective on a macroscopic scale as shown in optical micrographs in Fig. 6. The exposed, but unpassivated, BP surface (Fig. 6a) is roughened with adsorbed droplets[23], but the passivated surface (Fig. 6b) retains a mirror-like appearance for a period of several months and, apart from a few meandering cracks which appear dark, very few features which give rise to optical contrast. It has been suggested that the formation of droplets on the BP surface is due to the strongly

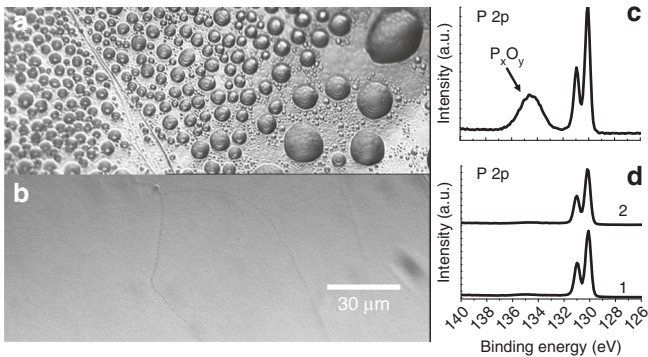

**Fig. 6** Stabilisation of black phosphorus surface by melamine cyanurate. Optical micrographs and corresponding P 2p regions of XPS spectra of the unterminated (**a**, **c**) and CA.M-terminated (**b**, **d**) surface of BP crystals after exposure to ambient conditions. Image **a** was acquired a week after exfoliation. Image **b** was taken 3 months after the CA.M was deposited. Panel **c** shows XPS spectrum that was acquired for an unterminated sample after exposure to ambient conditions for a week; the broad peak at 134.5 eV shows evidence for phosphorus oxides. Panel **d** shows XPS spectra acquired for CA.M-terminated samples within a few hours of preparation (curve 1) and 20 days after preparation (curve 2); there is no evidence for oxide formation on the CA.M-terminated surfaces

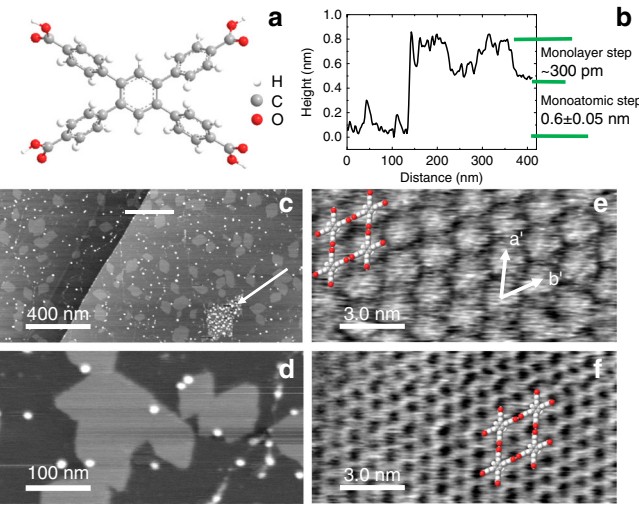

**Fig. 7** Islands of 1,2,4,5-tetrakis(4-carboxyphenyl)benzene on the melamine cyanurate-terminated black phosphorus surface. **a** Structure of 1,2,4,5-tetrakis(4-carboxyphenyl)benzene (TCPB); **b** Profile showing heights of monolayer TCPB islands and a BP terrace step as indicated in image **c**. **c**, **d** 2 μm × 1 μm and 500 nm × 250 nm, respectively, tapping mode AFM scans of TCPB islands; arrow in **c** shows an unterminated BP area which has degraded. These images were taken with Multi75Al-G probes using the fundamental resonance of 59.54 kHz; **e** Tapping mode AFM of a TCPB island showing individual molecules with lattice vectors **a'** and **b'** and schematic of hydrogen-bonded array overlaid. **f** Tapping mode AFM of the underlying CA.M network with TCPB lattice vectors and schematics overlaid. Images **e**, **f** were taken with Multi75Al-G probe using third harmonic of 1.107 Mhz

hydrophilic nature of the oxidised BP surface[55]. The obvious suppression of droplet formation in Fig. 6b, and the clear contrast with the behaviour of the unterminated surface (Fig. 6a), provide very strong evidence of both the effectiveness of this route for the surface passivation of BP and also its efficacy over large length scales.

The protective effect of the CA.M monolayer is further supported by X-ray photoelectron spectroscopy (XPS) measurements acquired from coated (Fig. 6d) and uncoated (Fig. 6c) surfaces of BP. For all samples we observed a clear P 2p doublet with peaks at 131.0 eV and 130.1 eV in excellent agreement with published data[23, 56]. The spectrum of an unterminated sample exposed to atmosphere for a week showed an additional broad peak centred at 134.5 eV related to phosphorus atoms bonded to oxygen[23]. XPS spectra of coated samples were acquired immediately after CA.M deposition (curve 1) and then after 20 days under ambient conditions (curve 2). The peak associated with oxidised phosphorus is absent from the data in Fig. 6d (apart from a very low intensity feature in curve 1) and the XPS spectrum is essentially unchanged for a sample exposed to ambient conditions for 20 days (curve 2). Our data agree well with published spectra[23] and there are no additional peaks which would imply that the adsorption of the CA.M monolayer results in a change of the BP surface bonding. We have also acquired Raman spectra of the clean surface, and CA.M-terminated surfaces, both immediately after molecular deposition, and after storage under ambient conditions for 1 month. These results are included in the Supplementary Information (Supplementary Fig. 4) and show a ratio of the intensities of the peaks associated with the $A^1_g$ and $A^2_g$ modes[31, 55, 57–61] of ~0.4, consistent with an oxide-free surface[31, 57]; this value is unchanged for the CA.M-terminated sample exposed to the atmosphere for a month consistent with the passivating effect discussed above.

We have also investigated whether, in addition to acting as a passivating layer, CA.M can also act as a stable interfacial layer for the adsorption of further molecular layers. Specifically, we have investigated the deposition of TCPB on CA.M/BP (Fig. 7) by immersion of the latter into a solution of TCPB in EtOH (see Methods). The TCPB molecule has four carboxyphenyl groups symmetrically attached to a central benzene core (Fig. 7a; see

Methods for details on the synthesis of TCPB). An AFM scan (2 μm × 1 μm) of the surface after the deposition of TCPB (Fig. 7c) reveals a topography similar to CA.M/BP apart from islands of TCPB with typical lateral dimensions of ~100 nm which are evenly distributed across the surface. The step height of a single island is ~0.3 nm (see Fig. 7b).

Higher-resolution scans (Fig. 7d) show that the TCPB islands have facetted edges and, in images with molecular resolution (Fig. 7e), we resolve a rhombic arrangement of TCPB molecules within the islands with lattice vectors, **a'** and **b'**, which are aligned with those of the CA.M network. The alignment is determined by AFM-initiated removal of TCPB using a slightly elevated setpoint in tapping mode to expose the underlying CA.M network (Fig. 7f), analogous to the removal of CA.M to reveal BP described above. From these images we determine a commensurate arrangement of TCPB on CA.M; the vectors **a'** and **b'** have a magnitude, within experimental error equal to $2a_{CA.M}$. In this arrangement, which is overlaid on Fig. 7e, f, the in-plane bonding between TCPB molecules is stabilised by double hydrogen bonds between neighbouring carboxyl groups. The stability of such an arrangement in the gas phase has been confirmed by numerical calculation of optimised structures at the semiempirical level using AM1 and RM1 procedures as implemented in the Firefly QC package (v 8.0.1)[62, 63]. These methods predict very similar rhombic unit cell dimensions of 1.86 nm (AM1) and 1.88 nm (RM1). These values are smaller than the 1.98 nm by ~0.1 nm observed in experiment, implying that the interaction with the CA.M layer leads to a slightly strained configuration.

It is clear that that the CA.M network is sufficiently stable to withstand the deposition of additional molecular layers and exhibits the same structural parameters, including a linear moiré pattern, providing further confirmation that the CA.M/BP interface is stable under both ambient conditions and under an

ethanolic solution of TCPB (note that immersion times of 20 min were required to deposit TCPB; the unterminated BP surface is not stable in ethanol even for much shorter times as shown for a control sample in Supplementary Information (Supplementary Fig. 5)), confirming that this approach provides a versatile method for the surface functionalisation of BP.

## Discussion

Overall, our results show clear evidence that CA.M passivates BP and provides a stable interfacial layer for further molecular deposition. The clearest evidence comes from the high-resolution AFM images which reveal the rows of P atoms when the CA.M monolayer is removed. The optical images, particularly the striking contrast between the terminated and unterminated surfaces in Fig. 6, also provide strong support that oxidation of BP is macroscopically inhibited by CA.M. Moreover, we emphasise that the fact that molecular monolayers can be formed on BP at all provides very strong evidence of a high-quality flat interface. To date, the solution deposition of supramolecular arrays has been demonstrated only on a limited range of sample substrates such as graphite, boron nitride, $MoS_2$ and Au(111) which provide stable, ultraflat, defect-free surfaces[34, 64, 65]. Our work clearly demonstrates that BP is also suitably flat and homogeneous to support two-dimensional molecular self-assembly. Furthermore, we observe a moiré pattern at the BP/CA.M interface, and the alignment of the CA.M structure with the phosphorus rows of the BP surface. These observations show that there is an epitaxial arrangement between the CA.M and the crystallographic directions of the BP confirming that the molecular layer is formed directly on the BP surface rather than an oxidised intermediate layer; our XPS and Raman data also support this conclusion.

A likely mechanism for the protective effect of a CA.M monolayer is that it simply forms an effective physical barrier between BP and the ambient environment. We note the strong bonding provided by the nine intermolecular hydrogen bonds (each hydrogen bond has a calculated[44] binding energy of ~0.26 eV) between a CA molecule and its three M neighbouring molecules, and the very limited solubility of melamine cyanurate[66], the bulk analogue of CA.M, in water. We also suggest that CA.M provides more effective passivation than TMA partly due to the more open nanoporous structure of the TMA 'chickenwire' supramolecular arrangement[39], and partly due to the differences in morphology of the CA.M and TMA monolayers. Note that the CA.M islands grow in a single orientation due to the epitaxial match and alignment with the BP crystal structure. In contrast TMA grows with multiple orientations and, consequently, even if a complete coverage could be formed, a much higher density of defects would be expected at the boundaries between molecular islands with different orientations.

We have explored these differences further by extending our study of molecular self-assembly on BP to include a third molecule, tetra(carboxyphenyl)porphyrin (TCPP). These new results are shown in Supplementary Fig. 6; we see islands of molecules on the surface in a similar arrangement to that previously reported for the same molecule[67] on boron nitride. In terms of passivating the BP surface we find a very similar outcome to TMA; the exposed BP regions degrade rapidly while the TCPP provide some short-term protection to the areas of the BP where it is adsorbed. TCPP, like TMA, has a porous supramolecular arrangement[67]. Furthermore, a highly significant technical issue is that it has not been possible to form a complete monolayer of TMA or TCPP on the BP surface. The growth rate of these materials is low and we are unable to increase this by extending the growth time since the unprotected BP has a limited lifetime in the relevant solvent, ethanol (EtOH; see Supplementary Fig. 5 for

images which show degradation of the BP surface in EtOH). In contrast, CA.M grows more rapidly and, due to the unique epitaxial orientational alignment, forms much bigger islands with lateral dimensions on the micron scale. This results in a lower defect density and a more complete coverage resulting in much more effective protection of the underlying BP surface. Note that the sample discussed here subsequently degraded and had a lifetime of more than 1 month but less than 2 months; the CA.M layer on a second sample subjected to lifetime tests remained intact for over 3 months (Supplementary Fig. 7) and we have prepared many samples for experiments other than lifetime tests which remained stable for the duration of the relevant investigation, typically several weeks.

In summary, our work demonstrates that a single layer of CA.M can successfully passivate the surface of BP and preserve it intact for over a month. This facile approach of depositing a passivating organic monolayer stabilised by in-plane noncovalent bonding could be extended to the protection of other two-dimensional materials with air-sensitive atomically flat surfaces, and is likely compatible with other solvents, molecules and possibly other coupling mechanisms such as halogen bonding[68]. Thus, it could provide a comprehensive supramolecular platform to preserve such materials under ambient conditions, in solvents (relevant to inks), and when using them in van der Waals heterostructures and devices. Additionally, we have demonstrated that the CA.M network provides a stable supramolecular platform for the sequential solution deposition of a second layer of molecules, potentially facilitating the solvent deposition of many other functional molecules and nanostructures such as quantum dots. The supramolecular arrays described here thus provide a route to both the passivation and functionalisation of the BP.

## Methods

**AFM imaging.** Images of the surface were acquired using a Cypher AFM (Oxford Asylum) in both AC (AM-AC and FM-AC) and contact modes in conjunction with Multi75Al-G cantilevers with a nominal spring constant of ~2.8 Nm$^{-1}$ and resonant frequency of ~75 kHz (Budget Sensor, Bulgaria). We have also used NuNano Scout 350 probes with a nominal spring constant of 42 Nm$^{-1}$ and resonant frequency of ~350 kHz (Nu Nano Ltd, Bristol, UK) and high-frequency Arrow UHF probes (NanoWorld AG, Neuchâtel, Switzerland) with a mean resonant frequency of ~ 2MHz. The spring constants and optical sensitivities of the levers were determined (to estimate the amplitude of oscillations) via the thermal tuning option as implemented in the Asylum Research software package. The optical sensitivity, which was confirmed by acquiring force curves on a hard surface (HOPG), was 65 ± 3 nmV$^{-1}$ for Multi75Al-G and 9 ± 1 nmV$^{-1}$ for Arrow UHF probes. The respective spring constants were determined to be 2.3 ± 0.1 Nm$^{-1}$ and 8.9 ± 0.1 Nm$^{-1}$. We have also used Multi75Al-G probes at higher resonant frequencies ~451 kHz (second harmonic) and ~1280 kHz (third harmonic). The exact mode and conditions are indicated in the relevant figure captions. AFM images were analysed and extracted with WSxM software (WSxM solutions, www.wsxm. es)[69].

**Molecular deposition.** TMA, CA and M were purchased from Sigma Aldrich. TMA was deposited from a 190 μM ethanolic solution and the CA.M complex from 8 μM aqueous solution (see protocol described previously[53]). TMA was deposited via immersion of BP for 180 s in a solution of TMA in EtOH. The deposition time to form an ~1 monolayer of CA.M was 60 s. The deposition was performed directly after the top layers of BP were removed by exfoliation under ambient conditions. After deposition the surface was blown dry with nitrogen for ~1 min. TCPB (synthesis described below) was deposited by immersion in a 90 μM solution of TCPB in EtOH. The deposition time was 20 min and the surface was dried in a N$_2$ stream. The samples were then immediately transferred to AFM for imaging. All samples were routinely stored in standard unsealed plastic boxes in the lab. TCPP was synthesised and deposited using protocols described previously[67].

**Characterisation using XPS and Raman spectroscopy.** Samples were analysed using the Kratos AXIS ULTRA with a mono-chromated Al kα X-ray source (1486.6 eV) operated at 10 mA emission current and 12 kV anode potential (120 W). Hybrid (electrostatic and magnetic) lens mode was used measuring a sample area of ~0.5 mm$^2$. The analysis chamber pressure was better than 5 × 10$^{-9}$ mbar. A low-resolution 'wide/survey' scan was acquired over the full energy range,

1400 to 5 eV binding energy, for 20 min at pass energy 80 eV, step 0.5 eV. High-resolution spectra at pass energy 20 eV with step of 0.1 eV, sweep times of 10 min each were also acquired for photoelectron peaks from the detected elements. Casaxps (version 2.3.18dev1.0x) software was used for quantification and spectral modelling. Raman spectra were acquired using a Horiba LABRAM confocal microscope/spectrometer fitted with 532 nm laser source. Spectra were acquired from BP crystals with a 50x objective corresponding to spot size of 1.1 μm$^2$ and power of 4.0 μW. Neutral density filters were used to reduce the incident intensity to the specified level.

**Synthesis of TCPB**. The molecule TCPB was synthesised using a Suzuki coupling reaction. 1,2,4,5-Tetrabromobenzene (2.00 g, 5.08 mmol), 4-boronic acid phthalic ethylester (6.00 g, 31.0 mmol) and K$_2$CO$_3$ (5.60 g, 40.6 mmol) were mixed in a mixture of toluene (60 ml), ethanol (30 ml) and water (30 ml), and the mixture de-aerated under Ar for 15 min. [Pd(PPh$_3$)$_4$] (0.20 g, 0.088 mmol) was added to the reaction mixture with stirring, and the mixture heated to 95 ℃ for overnight under Ar. The resultant mixture was evaporated to dryness and taken up in CHCl$_3$ which had been dried over MgSO$_4$. The CHCl$_3$ solution was evaporated to dryness and the residue washed briefly with EtOH (10 ml). The resulting crude product (mainly 1,2,4,5-tetrakis(4-carboxyethylester phenyl)benzene) was hydrolysed by refluxing the tetraester in 2 M aqueous NaOH followed by acidification with 37% HCl to to give the product as a white precipitate. Yield: 2.41 g, 85%.$^1$H nuclear magnetic resonance (DMSO-d$_6$, 300 MHz), Et$_4$L: 7.94 (d, J = 8 Hz, 8H), 7.57 (s, 2H), 7.30 (d, J = 8 Hz, 8H), 4.40 (q, 8H), 1.45 (t, 12H). H$_4$L: 7.84 (d, J = 8 Hz, 8H), 7.57 (s, 2H), 7.35 (d, J = 8 Hz, 8H). Elemental analyses (% calc/found) for C$_{34}$O$_8$H$_{22}$: C 73.1/73.3, H: 3.97/4.10.

**Data availability**. The raw data for the AFM images, XPS and Raman measurements may be accessed through the University of Nottingham Research Data Management Repository at http://dx.doi.org/10.17639/nott.316.

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

## Acknowledgements

This work was supported by the Engineering and Physical Sciences Research Council (grant numbers EP/N033906/1, EP/I011870/2, EP/K005138/1 and EP/K01773X/1); the Leverhulme Trust (grant number RPG-2016-104); and the European Research Council (grant number AdG 226593). Nottingham Nanoscale and Microscale Research Centre enabled access to the Raman and XPS instruments

## Author contributions

V.V.K., I.G.T. and P.H.B. conceived the experimental project; the AFM and Raman experiments were carried out by V.V.K. and the XPS measurements by E.F.S.; the BP crystals were grown by I.G.T. and R.H. and the TCPB and TCPP molecules were synthesised by S.Y., M.S., N.R.C. and L.Y.; V.V.K. and P.H.B. analysed the data and wrote the paper with revisions and comments from all authors.

## Additional information

**Competing interests:** The authors declare no competing financial interests.

