## [Peer Review File · Nature Communications]

Reviewers' comments:

Reviewer #1 (Remarks to the Author):

The author has made the following claims in the submitted manuscript:

1. BP supports the formation of supramolecular networks stabilised by in-plane hydrogen bonding.
2. These monolayers of supramolecules can passivate the BP surface and prevent degradation.
3. The supramolecules of 'CA.M' were deposited by dipping into solutions and further deposition TCPB were deposited on top of CA.M to show further molecular growth over passivated BP.
4. High Resolution AFM was used throughout to study and identify the surface crystal parameters and patterns and co-related to known patterns and lattice constants.

Comments:

1. The authors have presented a good and reasonable solution dispersion technique to use the nature of BP molecules to form supramolecules by combining with TMA and CM.A molecules and using their novel growth technique, which they have referenced to their recent work of growing similar molecules over hBN. Since, hBN has a similar structure to BP, their approach for BP is appreciation worthy.

2. They have supported their claim well by using high resolution AFM images and have identified the correct crystal parameters to identify BP and other materials and have similar identified Moiré patterns in CM.A to prove that these materials have been deposited over BP and form their suggested chain structures (formed by H-bonding) which they have shown with schematic diagrams. I agree with their claim that these materials have been deposited over BP and might also exist in the H-bonded chains as they have shown. There is ample proof given for that. (AFM imaging).

3. They have also used (fig. 3) an increased tip force to scratch the top layer to study the relative orientation of CA.M molecules with BP and have suggested a periodic overlay along one of the axes. This technique is good and they have justified it well enough by citing previously done STM measurements.

4. The use of these properties to further deposit TCPB molecules over CA.M has been demonstrated to show that stable supramolecular heterostructures can also be made over this passivation layer. I agree to this claim, as they have shown successful growth and identified the material with AFM imaging.

5. Now, the authors claim that BP underneath CA.M can be passivated or prevented from oxidation for up to 3 months, but they have only imaged the entire part (including both coated and uncoated part of BP), they have not given any proof/discussion that BP underneath is preserved. They could have used higher tip force to scratch through CA.M after three months (as done in Fig. 3) to see whether BP still exists with similar crystal parameters or it has been oxidised too? There is no measurement of the surface under the coating, which can support the claim that the BP underneath is still intact. Of course, the uncoated part will oxidise, which we can see, but there is nothing to show that the BP underneath the coating is still not oxidised. I have my reservations about that.

6. To make a claim that BP can be protected, the authors have not done any other experimental measurement suggesting BP still retains its original properties. Assuming, that the authors do not feel the need for such an experiment, they have not cited any relevant references claiming that interaction of such chemical components will not change the actual properties of BP. As we have observed in such passivation attempts earlier, that some sort of defects or other chemical defects are observed in BP. There is no such experimental or theoretical discussion/suggestion.

7. The authors have not commented anything about the effect of EtOH solution, in which the

samples were prepared. As suggested in the following reference;
<http://pubs.acs.org/doi/abs/10.1021/acsnano.5b02683> , It affects the cleavage of BP layers. The authors have not cited any reference or discussed the impact of the solutions used for chemical formation of supramolecules.

8. Several important statements in the manuscript are not referenced at all, which is not professional writing. For example, as follows;

“In this paper we explore a new route to the solution of this problem through an investigation of the compatibility of BP with the formation of supramolecular networks which have monolayer thickness and are stabilised by non-covalent in-plane interactions, specifically hydrogen bonding. This is motivated by the extensive literature of the formation of similar supramolecular networks on graphite, which has some structural similarity to BP but is much less reactive”.

“We also note that melamine is an organic base and as such reacts readily with acids forming ionic interactions which are typically about 10x times stronger than hydrogen bonds. This is an important consideration giving that various phosphoric acids are the usual chemical species formed during the oxidation process of the BP surface. Hence, if any of these acidic species were present on the surface then melamine would preferentially react with them forming salts which would lead, depending on the surface concentration, to complete or partial disruption of the CA.M network”.

In conclusion, I would like to say that the author has stated a good technique to grow a set of molecular chains over BP, but the author's claim to protect and passivate BP seems farfetched based on these observations. There must be enough experimental evidence to support the claim that these supramolecules actually protect BP from oxidation.

Reviewer #2 (Remarks to the Author):

The manuscript "back to black..." by Korolkov and co-workers reports highly original, interesting work and is generally a pleasure to read.

I recommend accepting it subject to minor revision.

Some minor comments.

Top of page 7, the description preceding Figure 3 is quite technical. I suggest rephrasing it and simplifying.

On page 7 (bottom) the authors state that they have recently "shown". The supporting reference however is only submitted. I suggest replacing "shown" with "observed".

page 11, I imagine that "giving" at the bottom should be "given"?

Several sentences start with Fig.; I believe a sentence should not start with an abbreviation.

For future work, the authors may consider trying to use halogen bonding as an alternative non-covalent type of interaction?

Reviewer #3 (Remarks to the Author):

The work of Korolkov et al. targets the formation of supramolecular networks stabilised by hydrogen bonding on the surface of BP crystals analysed by high-resolution AFM microscopy. The authors claim that the formation of hexagonal supramolecular networks consisting of trimesic acid and melamine cyanurate prevents the degradation of the BP surface for more than three months. Moreover, they used this as a platform for the subsequent deposition of 1,2,4,5-tetrakis(4-carboxyphenyl)benzene forming supramolecular heterostructures.

The main findings are:

- 1) Supramolecular networks prepared by solution deposition and stabilised by hydrogen bonding can be formed on the surface of BP.
- 2) The networks prepared with trimesic acid and melamine cyanurate can passivate the BP surface for more than three months, and act as platforms for further sequential deposition.

The paper presents results about three different commercially available molecules, namely trimesic acid (TMA), melamine (M) and cyanuric acid (CA), and their supramolecular assembly on BP crystals followed by high-resolution AFM. The formation of similar supramolecular networks with melamine cyanurate has been previously developed in graphene and related 2D materials, and this paper represents a nice example of the extrapolation to BP.

In overall, this is the first study on the supramolecular disposition of molecules on the surface of BP studied by high-resolution AFM. However, the authors claim a stabilization of the surface without taking into account some important points. Some crucial points should be addressed before considering its publication:

The introduction suffers from a general lack of precision in the citations. The first reports on the BP exfoliation are not properly cited (see ref. 5 and 12 of the manuscript).

Moreover, conceptually this article is not completely novel, as the non-covalent functionalization (and passivation) of BP has been already experimentally reported and thoroughly analysed by theoretical calculations (indeed, they omitted any reference to these previous reports): *Angew. Chem. Int. Ed.* 2016, 55, 14557–1456; *Adv. Mater.* 2017, 1603990. Concerning the mechanism of oxidation, I encourage the authors to consider some recent works (*Chem. Mater.*, 2016, 28, 8330–8339; *Nat Commun.*, 2015, 6, 8563) and modify the introduction accordingly.

With respect to the passivation effect, the authors developed the exfoliation procedure under environmental conditions without taking into account the spontaneous oxide layer formation on the surface of the BP flakes. In this sense, the absence of a phosphorous oxide layer should be demonstrated; otherwise the system under study will consist of oxidized BP, thus affecting the interaction with the molecules at the interface.

Moreover, the authors omitted any reference to the thickness, lateral dimensions of the flakes and light exposure, a matter of utmost importance in the degradation of BP. In this sense, the authors only provided evidences of passivation in bulk BP. A similar study on well-defined few-layers BP (including different samples) is necessary for claiming the stabilisation of BP.

According to Fig. 2 supramolecular layers of TMA can be formed on BP, but creating inhomogeneous islands that precludes the stabilisation of BP. Could you provide some hints on the limitations of this route compared to the melamine-cyanurate one? What is the influence of nitrogen groups on the organisation of the networks and stabilization of BP?

Moreover, the mechanism underlying the protective effect is unclear and requires further experiments. As reported elsewhere, the presence of melamine in graphene results on a charge transfer from the graphene to the melamine. Similar charge transfer behaviour has been reported for BP in the presence of electron withdrawing molecules like TCNQ. A thorough analysis of this phenomenon should be performed, including DFT calculations. In addition, the acid-base discussion of the page 11 must be developed in more detail, including experimental evidences. Some additional characterization of the functionalized surfaces should be provided including XPS and Raman spectroscopy.

In overall, I consider this work interesting and could deserve publication after addressing all the aforementioned critical points.

Response to Referees

Back to black: supramolecular networks stabilise and functionalise black phosphorus NCOMMS-17-10963

Referee 1

The referee correctly summarises our main claims and then continues to offer the following comments (in italics) to which we respond.

Comments:

1. The authors have presented a good and reasonable solution dispersion technique to use the nature of BP molecules to form supramolecules by combining with TMA and CM.A molecules and using their novel growth technique, which they have referenced to their recent work of growing similar molecules over hBN. Since, hBN has a similar structure to BP, their approach for BP is appreciation worthy.

We are pleased the referee recognizes the validity of our interpretation.

2. They have supported their claim well by using high resolution AFM images and have identified the correct crystal parameters to identify BP and other materials and have similar identified Moiré patterns in CM.A to prove that these materials have been deposited over BP and form their suggested chain structures (formed by H-bonding) which they have shown with schematic diagrams. I agree with their claim that these materials have been deposited over BP and might also exist in the H-bonded chains as they have shown. There is ample proof given for that. (AFM imaging).

This aspect of our work is also accepted by the referee.

3. They have also used (fig. 3) an increased tip force to scratch the top layer to study the relative orientation of CA.M molecules with BP and have suggested a periodic overlay along one of the axes. This technique is good and they have justified it well enough by citing previously done STM measurements.

The referee recognizes the validity of this approach to remove the molecular layer and verify that the underlying BP surface has remained protected.

4. The use of these properties to further deposit TCPB molecules over CA.M has been demonstrated to show that stable supramolecular heterostructures can also be made over this passivation layer. I agree to this claim, as they have shown successful growth and identified the material with AFM imaging.

The demonstration that the CA.M layer also provides a stable interface on which additional layers can be deposited is also accepted by the referee.

5. Now, the authors claim that BP underneath CA.M can be passivated or prevented from oxidation for up to 3 months, but they have only imaged the entire part (including both coated and un-coated part of BP), they have not given any proof/discussion that BP underneath is preserved. They could have used higher tip force to scratch through CA.M after three months (as done in Fig. 3) to see whether BP still exists with similar crystal parameters or it has been oxidised too? There is no measurement of the surface under the coating, which can support the claim that the BP underneath is still intact. Of course, the uncoated part will oxidise, which we can see, but there is nothing to show that the BP underneath the coating is still not oxidised. I have my reservations about that.

We have followed the referee's suggestion and repeated the scratching experiments. In order to do this we have had to prepare a new sample (and procure some additional BP substrate material). The new sample was prepared in early July and due to time constraints on the re-submission of our paper we have successfully reproduced the scratching experiment at weekly intervals up to just over a month and plan to continue these experiments at monthly intervals during the second refereeing cycle. The new results are included in Figure 5 (in fact the resolution in AFM in these new images surpasses that available in our original manuscript). Additional intermediate scratching experiments are included in a new Supplementary Information which we have added.

In the revised version we refer to the protection as being effective for a period of 'over a month'. In fact we believe that a protective effect even on this timescale represents a major step forward but we will also update the paper, if accepted, based on our ongoing checking by adding further test data over a longer timescale to the Supplementary Information.

6. To make a claim that BP can be protected, the authors have not done any other experimental measurement suggesting BP still retains its original properties. Assuming, that the authors do not feel the need for such an experiment, they have not cited any relevant references claiming that interaction of such chemical components will not change the actual properties of BP. As we have observed in such passivation attempts earlier, that some sort of defects or other chemical defects are observed in BP. There is no such experimental or theoretical discussion/suggestion.

In our revised paper we have added XPS results (to the main paper Figure 6) and Raman spectroscopy measurements (added to Supplementary Information). Relevant to this point the XPS results acquired for a sample which is terminated by CA.M show the same peaks as previously reported for BP with no evidence for additional peaks which would indicate a change of bonding of the near surface material (apart from a weak and broad feature due to oxidation which we attribute to the small areas of unterminated surface revealed in our AFM images). The XPS of the CA.M terminated sample remains unchanged on exposure to atmosphere. The Raman spectrum of a freshly prepared CA.M terminated BP surface is unchanged after a month providing further support for the protective effect. These results show that there is no evidence for chemical changes to the BP – although the strongest evidence comes from the scratching experiments. We have added several additional sections to our discussion related to these points.

7. The authors have not commented anything about the effect of EtOH solution, in which the samples were prepared. As suggested in the following reference; <http://pubs.acs.org/doi/abs/10.1021/acsnano.5b02683>, It affects the cleavage of BP layers. The authors have not cited any reference or discussed the impact of the solutions used for chemical formation of supramolecules.

The paper mentioned by the referee is focused on liquid phase exfoliation which is not the procedure used in our study. Following the referee's comment we have checked the effect of immersion of unterminated BP in EtOH. This leads to a roughened surface similar to that observed after extended exposure to atmosphere. This result has been added to the Supplementary Information and is mentioned in a discussion related to the mechanism for protection (see comments to Referee 3). We also emphasise that the preservation of the CA.M layer after immersion in the TCPB solution provides direct evidence that the surface layers have not been exfoliated by immersion, although this would anyway be, in our view, a highly unlikely outcome.

8. Several important statements in the manuscript are not referenced at all, which is not professional writing. For example, as follows;

"In this paper we explore a new route to the solution of this problem through an investigation of the

compatibility of BP with the formation of supramolecular networks which have monolayer thickness and are stabilised by non-covalent in-plane interactions, specifically hydrogen bonding. This is motivated by the extensive literature of the formation of similar supramolecular networks on graphite, which has some structural similarity to BP but is much less reactive”.

We thank the referee for pointing this out and presume they refer to the absence of a general citation to work on molecular adsorption on graphene. We have added references to review articles on this topic.

“We also note that melamine is an organic base and as such reacts readily with acids forming ionic interactions which are typically about 10x times stronger than hydrogen bonds. This is an important consideration giving that various phosphoric acids are the usual chemical species formed during the oxidation process of the BP surface. Hence, if any of these acidic species were present on the surface then melamine would preferentially react with them forming salts which would lead, depending on the surface concentration, to complete or partial disruption of the CAM network”.

On re-evaluation of our paper we have removed this section since it refers to an effect which was not observed. It seems unnecessary, potentially confusing and not directly relevant to our results to speculate on why other outcomes are not observed.

In conclusion, I would like to say that the author has stated a good technique to grow a set of molecular chains over BP, but the author’s claim to protect and passivate BP seems farfetched based on these observations. There must be enough experimental evidence to support the claim that these supramolecules actually protect BP from oxidation.

In fact even in the first version of the manuscript the scratching experiments shown in Fig. 3a were performed on samples which had been exposed to the ambient for 2 days. Therefore these results already demonstrated a significant passivating effect and we find it difficult to reconcile this observation with the referee’s description that our claim is ‘farfetched’. We hope that the referee will accept that the additional supporting evidence provided primarily by the longer term scratching experiments demonstrates the protective effect convincingly.

Referee 2

This referee recommends publication following consideration for of several points which we address below:

Some minor comments.

Top of page 7, the description preceding Figure 3 is quite technical. I suggest rephrasing it and simplifying.

We have removed some of the technical points for what are relatively simple observations – that the TMA provides limited protection of the surface.

On page 7 (bottom) the authors state that they have recently "shown". The supporting reference however is only submitted. I suggest replacing "shown" with "observed".

We have modified the wording as suggested. The unpublished reference has now appeared in Nature Chemistry online and the relevant citation has been modified.

page 11, I imagine that "giving" at the bottom should be "given"?

This section has been removed (see comments to Referee 1).

Several sentences start with Fig.; I believe a sentence should not start with an abbreviation.

This has been changed as suggested.

For future work, the authors may consider trying to use halogen bonding as an alternative non-covalent type of interaction?

This is an interesting suggestion which we have included in our discussion section and added a relevant citation.

Referee 3

We respond to the referee below.

The work of Korolkov et al. targets the formation of supramolecular networks stabilised by hydrogen bonding on the surface of BP crystals analysed by high-resolution AFM microscopy. The authors claim that the formation of hexagonal supramolecular networks consisting of trimesic acid and melamine cyanurate prevents the degradation of the BP surface for more than three months. Moreover, they used this as a platform for the subsequent deposition of 1,2,4,5-tetrakis(4-carboxyphenyl)benzene forming supramolecular heterostructures.

The main findings are:

- 1) Supramolecular networks prepared by solution deposition and stabilised by hydrogen bonding can be formed on the surface of BP.*
- 2) The networks prepared with trimesic acid and melamine cyanurate can passivate the BP surface for more than three months, and act as platforms for further sequential deposition.*

The paper presents results about three different commercially available molecules, namely trimesic acid (TMA), melamine (M) and cyanuric acid (CA), and their supramolecular assembly on BP crystals followed by high-resolution AFM. The formation of similar supramolecular networks with melamine cyanurate has been previously developed in graphene and related 2D materials, and this paper represents a nice example of the extrapolation to BP.

In overall, this is the first study on the supramolecular disposition of molecules on the surface of BP studied by high-resolution AFM. However, the authors claim a stabilization of the surface without taking into account some important points. Some crucial points should be addressed before considering its publication:

We agree with this positive summary of the main points of our paper and note that the referee appreciates that the solution deposition of any sort of supramolecular network on BP is a significant step forward. Coupled with the demonstration of passivation and the role as a platform for further deposition we argue strongly that our paper represents a major advance which brings together the sub-fields of 2D supramolecular organization and BP. We respond to the points raised by the referee below.

The introduction suffers from a general lack of precision in the citations. The first reports on the BP

exfoliation are not properly cited (see ref. 5 and 12 of the manuscript).

Moreover, conceptually this article is not completely novel, as the non-covalent functionalization (and passivation) of BP has been already experimentally reported and thoroughly analysed by theoretical calculations (indeed, they omitted any reference to these previous reports): Angew. Chem. Int. Ed. 2016, 55, 14557–1456; Adv. Mater. 2017, 1603990. Concerning the mechanism of oxidation, I encourage the authors to consider some recent works (Chem. Mater., 2016, 28, 8330–8339; Nat Commun., 2015, 6, 8563) and modify the introduction accordingly.

We have modified our introductory discussion and changed the references to the first report of exfoliation. We thank the referee for highlighting the additional papers related to molecular adsorption and oxidation and we have added citations to these papers and extended our discussion of prior work on molecular adsorption on BP.

With respect to the passivation effect, the authors developed the exfoliation procedure under environmental conditions without taking into account the spontaneous oxide layer formation on the surface of the BP flakes. In this sense, the absence of a phosphorous oxide layer should be demonstrated; otherwise the system under study will consist of oxidized BP, thus affecting the interaction with the molecules at the interface.

Related to a separate point below we have added XPS results which do not show evidence for oxidation of the CA.M monolayer terminated BP layers. These have been added to Figure 6 of the modified manuscript. We have also included in Supplementary Information the Raman spectra of the freshly exfoliated BP surface and, following Abellan et al., compared our results with Favron et al. who argue that the degree of oxidation can be determined from the measured ratio of the A_g^1 and A_g^2 peak intensities. According to these results the freshly exfoliated BP surface we use is free of oxidation (the measured ratio of peak intensities is 0.9). The ratio for the CA.M terminated surface is slightly lower (0.4) but still represents an oxide free surface according to the criterion applied by Abellan et al. and Favron et al.. Note that for the CA.M sample some local oxidation occurs at the gaps in coverage as revealed in AFM; this accounts, at least in part, for the reduction in the peak intensity ratio.

We emphasise that the most direct evidence for an oxide free surface comes from; (i) our AFM results; (ii) the very obvious difference in the optical appearance of the surfaces in Figure 6. We have provided a much more extensive discussion of these points in the revised paper. The protection of the surface is shown most directly by the scratching experiments. In particular, the observation of atomic rows on the BP surface both on the freshly exfoliated surface (Fig. 1) and the areas of the surface where the CA.M has been removed (Fig. 3) provides direct evidence of a clean BP surface. Note that there are no AFM images in the literature which come close to the resolution we present in our images and consequently this represents a new approach to investigating the surface structure of BP under ambient conditions.

Moreover, as we point out in our revised paper, the fact the molecular monolayers can be formed on BP at all provides very strong evidence of a high quality flat interface which is not consistent with the formation of an oxide layer. The referee may be aware that the solution deposition of supramolecular arrays is possible only on a limited range of sample substrates such as graphite, boron nitride, MoS_2 and Au(111) which provide stable, ultraflat, defect-free surfaces. Our work clearly demonstrates that BP may be added as a suitable substrate to this rather short list, providing direct evidence that the BP surface is suitably flat and homogeneous to support molecular self-assembly. Furthermore, we observe a moiré pattern at the BP/molecule interface, and the alignment of the CA.M structure with the crystallographic directions of BP. This shows that there is an epitaxial relationship between the molecular overlayer and the crystallographic directions of the BP which by

implication must form the interfacial layer directly below the molecular overlayer. We have added a new paragraph to emphasise these points in the revised manuscript.

Moreover, the authors omitted any reference to the thickness, lateral dimensions of the flakes and light exposure, a matter of utmost importance in the degradation of BP. In this sense, the authors only provided evidences of passivation in bulk BP. A similar study on well-defined few-layers BP (including different samples) is necessary for claiming the stabilisation of BP.

We fully acknowledge that our study is focused on bulk BP: at the outset of our study the solution deposition of this type of 2D hydrogen bonded networks on BP had not been studied at all, and we argue that our demonstration that their formation on bulk BP is possible and, furthermore, that they can provide passivation against atmospheric degradation and also an interfacial barrier for the growth of supramolecular heterostructures already represents very substantial progress. Note that this is only the second demonstration of a supramolecular heterostructure on any surface following our recent publication in Nature Chemistry (Ref 56). We agree that it will be interesting to extend this study to few-layer phosphorene and have started work towards this goal. There are various technical issues related to the cleanliness of the exfoliated flakes but we expect to resolve these very soon; nevertheless the current manuscript, already a very extensive piece of work, is focused on adsorption of supramolecular networks on bulk BP surfaces.

According to Fig. 2 supramolecular layers of TMA can be formed on BP, but creating inhomogeneous islands that precludes the stabilisation of BP. Could you provide some hints on the limitations of this route compared to the melamine-cyanurate one? What is the influence of nitrogen groups on the organisation of the networks and stabilization of BP? Moreover, the mechanism underlying the protective effect is unclear and requires further experiments. As reported elsewhere, the presence of melamine in graphene results on a charge transfer from the graphene to the melamine. Similar charge transfer behaviour has been reported for BP in the presence of electron withdrawing molecules like TCNQ. A thorough analysis of this phenomenon should be performed, including DFT calculations.

In response to the referee's suggestion we have performed more experiments including XPS and Raman, as discussed above, and also extended our study of molecular self-assembly to include a third molecule tetra(carboxyphenyl)porphyrin (TCPP). This molecule was chosen in part to respond to the question about the role of nitrogen (which is present in the porphyrin macrocycle). These new results are included in the Supplementary Information and show a very similar outcome to TMA; we see islands of molecules on the surface in a square arrangement similar to that previously reported for the same molecule on boron nitride. The exposed BP regions degrade rapidly while the TCPP provide some short term protection to the areas of the BP where it is adsorbed. Thus, this molecule shows rather similar behavior to the non-nitrogen containing TMA thus ruling out a simplistic explanation related to N atoms.

We have also responded to the referee's comment by greatly expanding our discussion of the differences between TMA and CA.M (now also including the behavior of TCPP). In this new section of the paper we argue that the protection mechanism is due to the CA.M forming a physical barrier between BP and the ambient environment. We note the strong bonding provided by the three intermolecular hydrogen bonds and the insolubility of the bulk analogue melamine cyanurate in water. An argument related to why CA.M works so much better than TMA and TCPP is also presented; we believe that this is because large CA.M islands grow in a single orientation due to the epitaxial match and alignment with the BP crystal structure. In comparison TMA and TCPP grow with multiple orientations and, as a consequence, even if a complete coverage could be formed, a much higher density of defects would be expected at the boundaries between molecular islands with different orientations. TMA and TCPP might also be expected to be less effective barrier materials

due to their higher porosity. Moreover, a highly significant technical issue is that it has simply not been possible to form a complete monolayer of TMA or TCPP. The growth rate of these materials is low (compared to other 2D substrates) and we are unable to increase this due to the limited solubility of these molecules, or by extending the growth time (since the BP has a limited lifetime in the relevant solvent EtOH – see additional control images added to the Supplementary Information). In contrast CA.M appears to grow more rapidly. We believe these differences account for the different efficacies of the various molecules.

Although the referee suggests DFT calculations we believe that our conclusions can be clearly argued without additional numerical calculations (note calculations of the structural parameters of TCPB where they were included in our original manuscript). In particular, whether such calculations showed that charge transfer either did or did not occur would not influence our interpretations of data, or our conclusions. Of course, the question of charge transfer is interesting although we are not aware of experimental results showing charge transfer from graphene to melamine (there are several theory papers) – or indeed on other substrates such as gold where this molecule has been extensively studied. Our experimental results to date, for example XPS, show no evidence for a change in the bonding of the surface P atoms.

..... *In addition, the acid-base discussion of the page 11 must be developed in more detail, including experimental evidences.*

As discussed above we have removed this section – see comments to Referee 1.

Some additional characterization of the functionalized surfaces should be provided including XPS and Raman spectroscopy.

We have performed these measurements and have included them in the main text and Supplementary Information as discussed above. They support our original findings.

In overall, I consider this work interesting and could deserve publication after addressing all the aforementioned critical points.

We hope the referee will agree that we have addressed the points which are most critical in relation to our stated conclusions. We acknowledge that further work in this area is highly desirable and anticipate that our work will provide a stimulus for this activity.

REVIEWERS' COMMENTS:

Reviewer #1 (Remarks to the Author):

- 1: The author has repeated the experiment based on the suggestion provided in Comment 5 of the earlier review. The results are acceptable and seem to support the claim of the author about the passivation of BP, which has been supported by XPS and Raman measurements.
- 2: I agree that this method can protect BP surface for a period of 'over a month' as the author has now added in the manuscript, with the claim that author will update the information before publication once further results of their ongoing measurements start to come in increasing the observation period.
- 3: Supporting experiments to rebut the Comments 6 and 7 are satisfactory and act as steps to further support the claim as was suggested. I agree with the discussion that they have added in the main text.
- 4: Referencing has been corrected as per suggestions in Comment 8.
- 5: The claim to passivate/protect BP can now be comprehended on the basis of further experimental evidence that has been added.

I recommend the article for publication.

1: The author has addresses several key concerns raised by Referee 3 by addressing the following:

- a. The author has now modified the introduction and added further new references stating the previous works that have been carried out in this field of passivation of BP.
- b. The authors have included new XPS and Raman spectroscopy measurements that support their argument against the effective oxidation of BP flakes. The referee agrees with their claims.
- c. Since, the author has now acknowledged that the BP surfaces that they used for passivation were bulk crystals not monolayers. The referee suggests that this should be highlighted in the abstract or in the main text somewhere. This is essential to differentiate this from other passivation studies on monolayer preservation.
(10.1038/ncomms10450)

The referee leaves this decision to the editor in charge.

d. The referee agrees with the other arguments and changes that have been made in compliance to the other referee comments and is satisfied with the answers to the rest of the questions.

e. The author has suggested that DFT calculations are not necessary to support their claim. But, in my opinion, in order to establish this passivation procedure as a proven phenomenon for other materials or even to be used for BP, as the authors claim in their conclusion, DFT calculations would be an essential step to study charge transfer/chemical effects of such passivation on BP.
But, I agree that the experiments presented here are satisfactory to support passivation of BP for 'a month'.

I recommend the article for publication.

Second Response to Referees

Supramolecular networks stabilise and functionalise black phosphorus NCOMMS-17-10963

Referee 3

We have included a more explicit statement in the main text that the work is performed on thick bulk-like BP substrates.